# Viral nucleoprotein antibodies activate TRIM21 and induce T cell immunity

Sarah L Caddy[1,2,*] , Marina Vaysburd[1], Guido Papa[1], Mark Wing[1], Kevin O'Connell[1], Diana Stoycheva[3], Stian Foss[4,5], Jan Terje Andersen[4,5], Annette Oxenius[3] & Leo C James[1,**]

## Abstract

**Nucleoprotein (N) is an immunodominant antigen in many enveloped virus infections. While the diagnostic value of anti-N antibodies is clear, their role in immunity is not. This is because while they are non-neutralising, they somehow clear infection by coronavirus, influenza and LCMV *in vivo*. Here, we show that anti-N immune protection is mediated by the cytosolic Fc receptor and E3 ubiquitin ligase TRIM21. Exploiting LCMV as a model system, we demonstrate that TRIM21 uses anti-N antibodies to target N for cytosolic degradation and generate cytotoxic T cells (CTLs) against N peptide. These CTLs rapidly eliminate N-peptide-displaying cells and drive efficient viral clearance. These results reveal a new mechanism of immune synergy between antibodies and T cells and highlights N as an important vaccine target.**

**Keywords** antibody; non-neutralising; nucleoprotein; TRIM21; virus
**Subject Category** Immunology
**The EMBO Journal (2021) 40: e106228**

## Introduction

Antibodies to surface glycoproteins (GP) on enveloped viruses (e.g. spike on coronavirus (Zost *et al*, 2020), hemagglutinin (HA) and neuraminidase (NA) on influenza (Burton *et al*, 2012) or glycoprotein (GP) on LCMV (Bruns *et al*, 1983)) are well known to neutralise infection. However, antibodies are also made against internal antigens, such as the nucleoprotein (N or NP) that is responsible for packaging viral genomes. Surprisingly, anti-N antibodies are often produced earlier during infection and reach a higher titre than their anti-glycoprotein counterparts. In acute LCMV infection, anti-N antibodies can be detected from as early as 4 days and remain higher in titre than anti-GP antibodies throughout the response (Battegay *et al*, 1993; Eschli *et al*, 2007). Crucially, while LCMV is almost entirely cleared within a few weeks (Moskophidis *et al*, 1993), a neutralising titre is often only

observed after several months and only when virus infection has become persistent (Fallet *et al*, 2020). This could be taken to suggest that antibodies are not important to resolve infection, but in mice with restricted antibody specificity (MD4; Straub *et al*, 2013) and T11μMT; Bergthaler *et al*, 2009), lacking functional B cells (JHT (Bergthaler *et al*, 2009)) or an ability to produce soluble antibodies (ΔIgMi Straub *et al*, 2013), viraemia persists for a month or more.

How the predominately anti-N, non-neutralising antibody response helps to clear infection remains a mystery, particularly as protection can be observed in both FcγR and complement C3/C4 knockouts (Bergthaler *et al*, 2009). Importantly, the ability of non-neutralising antibodies to control infection is common to many enveloped viruses including HIV (Mayr *et al*, 2017), HCMV (Bootz *et al*, 2017) and influenza (Sambhara *et al*, 2001; Carragher *et al*, 2008).

We hypothesised that anti-N antibodies may function by activating the cytosolic antibody receptor and E3 ubiquitin ligase TRIM21. TRIM21 is responsible for antibody-dependent intracellular neutralization or ADIN. It detects antibody-bound substrates inside the cell and mediates their rapid proteasomal degradation. When targeting an incoming virion, this results in a potent block to infection (Mallery *et al*, 2010). When targeting an individual protein, this causes its selective depletion and is the basis for the technology Trim-Away (Clift *et al*, 2017). We therefore sought to determine whether TRIM21 could interact with anti-N antibodies and provide a mechanism to explain the ability of non-neutralising antibodies to protect against enveloped viruses. Using LCMV as a model enveloped virus, here we show that non-neutralising anti-N antibodies provide potent immune protection by activating the cytosolic Fc receptor TRIM21. This is achieved by driving N protein antigen presentation and stimulating cytotoxic T cell killing.

## Results

### TRIM21 protects against LCMV infection in naïve mice

To investigate the role of TRIM21 in LCMV infection, we challenged naïve wild-type (WT) and TRIM21 knockout (KO) mice with a high

---

1   MRC Laboratory of Molecular Biology, Cambridge, UK
2   CITIID, Department of Medicine, University of Cambridge, Cambridge, UK
3   Department of Biology, Institute of Microbiology, ETH Zurich, Zurich, Switzerland
4   Department of Immunology, University of Oslo and Oslo University Hospital Rikshospitalet, Oslo, Norway
5   Institute of Clinical Medicine and Department of Pharmacology, University of Oslo and Oslo University Hospital, Oslo, Norway
    *Corresponding author. Tel: +44 1223 767049; E-mail: slc50@cam.ac.uk
    **Corresponding author. Tel: +44 1223 267162; E-mail: lcj@mrc-lmb.cam.ac.uk

dose of the isolate clone 13. A dramatic weight loss was observed from day 5 that was identical in both genotypes (Fig 1A). On day 8, WT animals began to regain weight and had fully recovered by day 12. In contrast, KO animals continued to lose weight throughout the observation period with 4/6 reaching end point by day 10 and the remaining mice succumbing to infection by day 12 (Fig 1B). A sharp divergence in clinical progression that is only apparent from the second week of infection is consistent with the onset of adaptive immunity and so we measured the antibody responses against viral nucleoprotein (N) and glycoprotein (GP). There was a detectable anti-N IgG response in WTs from day 6 that had rapidly increased by day 8, whereas IgG antibodies targeting the GP protein were only detected from day 15 (Fig 1C), in agreement with previously published data (Eschli et al, 2007). A similarly rapid increase in the anti-N response was observed in KO animals, suggesting that their sensitivity to LCMV infection is not due to a difference in antibody titres (Fig 1D).

Characterisation of the N-specific antibodies generated by day 8 showed that these were unable to neutralise LCMV in wild-type (WT) murine embryonic fibroblasts (MEFs), as expected (Fig 1E).

However, when the same sera were delivered directly into the cytosol of WT MEFs by electroporation prior to viral challenge, potent neutralisation was observed (Fig 1F). To test whether this intracellular neutralisation was dependent upon TRIM21, we electroporated WT and KO MEFs with the N-specific mAb KL53 (Zeller et al, 1988). There was a significant loss of neutralisation in KO cells over a wide-range of anti-N antibody concentrations (Fig 1G). TRIM21 has been shown to prevent adenoviral infection by catalysing ubiquitination and causing rapid proteasomal degradation of the structural protein hexon (Mallery et al, 2010). To test whether LCMV N protein is a substrate for TRIM21 degradation, we electroporated antibody:antigen complex composed of KL53 and recombinant N protein into WT and KO MEFs. Immunoblotting for N protein after 3 h showed depletion in WT cells but not KO (Fig 1H). Moreover, when recombinant KL53 containing the non-TRIM21 binding mutation H433A (James et al, 2007) was used, N depletion was also ablated in WTs. Overall, this suggests that after LCMV infection, TRIM21 knockout mice begin to diverge from their WT counterparts once an anti-N antibody response capable of mediating N degradation and intracellular neutralisation is made.

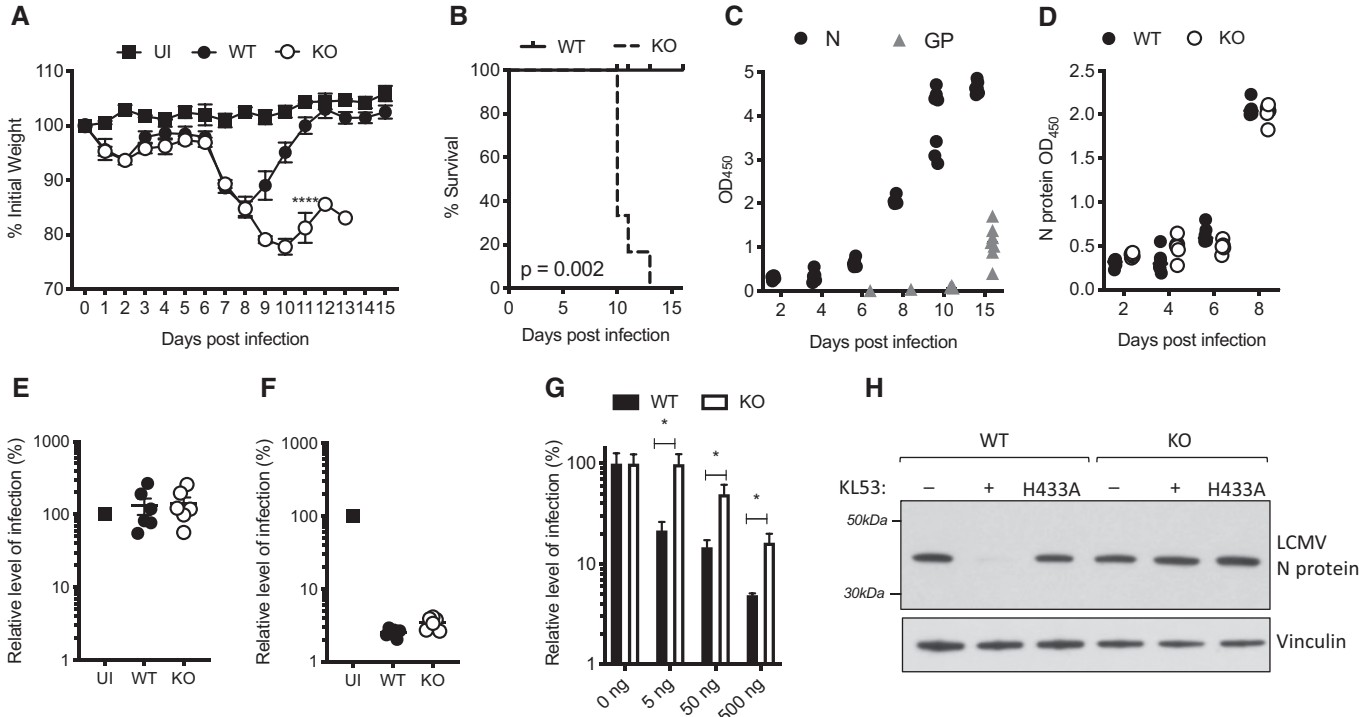

**Figure 1. TRIM21 uses anti-N antibodies to protect against LCMV infection.**

A–F    Wild-type (WT) and TRIM21 knockout (KO) mice were infected with $10^5$ FFU LCMV clone 13 (6 mice per group) and 2 mice were uninfected (UI). Weights (A) and Kaplan–Meier survival (B) were compared over 15 days post infection (pi). (C) Sera from WT mice was analysed for N protein (N) and glycoprotein (GP)-specific antibodies by ELISA, each data point corresponds to one mouse from the same experiment. (D) N-specific antibody ELISA was used to compare antibody responses in sera of WT and KO mice. (E, F) In vitro neutralisation experiments were performed with sera collected day 8pi from WT, KO and UI mice. (E) Sera was pre-incubated with LCMV, then the virus-serum mix was added to MEF cells and LCMV infection titre after 16 h was measured by FFA. (F) Sera was electroporated into MEFs, then cells were plated in triplicate and LCMV was added 4 h later. LCMV infection titre after 16 h was measured by FFA (three replicates).
G    Anti-N mAb KL53 was electroporated into WT and KO MEFs and subsequent LCMV infection titres were measured by FFA.
H    Anti-N mAb KL53 was co-electroporated with recombinant N protein into WT and KO MEFs, and immunoblotting for N was performed after 3 h. Electroporation of recombinant KL53 expressing the TRIM21 non-binding mutation H433A was unable to mediate N protein degradation.

Data information: All data are presented as mean with standard error, *$P < 0.05$, unpaired t-test.
Source data are available online for this figure.

    

## Anti-N antibodies mediate protection via TRIM21

To directly test whether anti-N antibodies mediate a protective response via TRIM21, we carried out a passive transfer experiment in which the anti-N mAb KL53 was administered intraperitoneally (IP) on days 1 and 3 post LCMV infection (pi). As previously shown, both genotypes behaved similarly during the first week of infection, with weight loss beginning around day 4/5 (Fig 2A). WT mice given KL53 began to recover earlier than naïve animals, at day 6, and by day 8 had regained their starting weight (Fig 2A). In contrast, KL53 provided no benefit to KOs and these animals continued to lose weight. At day 8, there were statistically significant differences between WT+/− KL53 ($P = 0.0009$), and WT compared with KO mice (no KL53 $P = 0.0022$, + KL53 $P = 0.0001$) but not between KO groups. To determine whether these patterns of weight loss are due to underlying differences in viral clearance, we analysed viral load on day 8pi in WT and KO mice +/− KL53 in the spleen, liver, lung and kidney (Fig 2B). Whereas administration of KL53 significantly decreased viral load in every organ analysed from WT mice, there was no significant difference in viral loads in KO mice.

## Non-neutralising antibodies and TRIM21 drive cytotoxic T cell activation

Anti-N antibodies and TRIM21 could be clearing viral infection either cell autonomously or by synergising with cellular immunity. Cytotoxic T cells (CTLs) in particular have been implicated in non-neutralising anti-N antibody protection for multiple viruses including influenza (Sambhara *et al*, 2001; Carragher *et al*, 2008) and LCMV (Richter & Oxenius, 2013; Straub *et al*, 2013). To test this, we performed passive transfer experiments in mice depleted of CTLs using anti-CD8 mAbs. Whereas previously we had observed that KL53 significantly reduced viral loads in multiple organs (Fig 2B), in mice depleted of CTLs, KL53 had no effect relative to control (Fig 3A). Confirmation of CTL depletion efficiency is shown in Fig EV1A. As CTL depletion abolished the protective effect of anti-N antibodies, we performed the converse experiment and asked whether anti-N antibodies promote induction of N-specific CTLs. Splenocytes were harvested at 8dpi in the presence or absence of passively transferred KL53 antibody. CTLs specific for the immunodominant N peptide (396–404) were significantly upregulated in the presence of KL53 ($P \leq 0.0001$) in WT mice, whereas there was

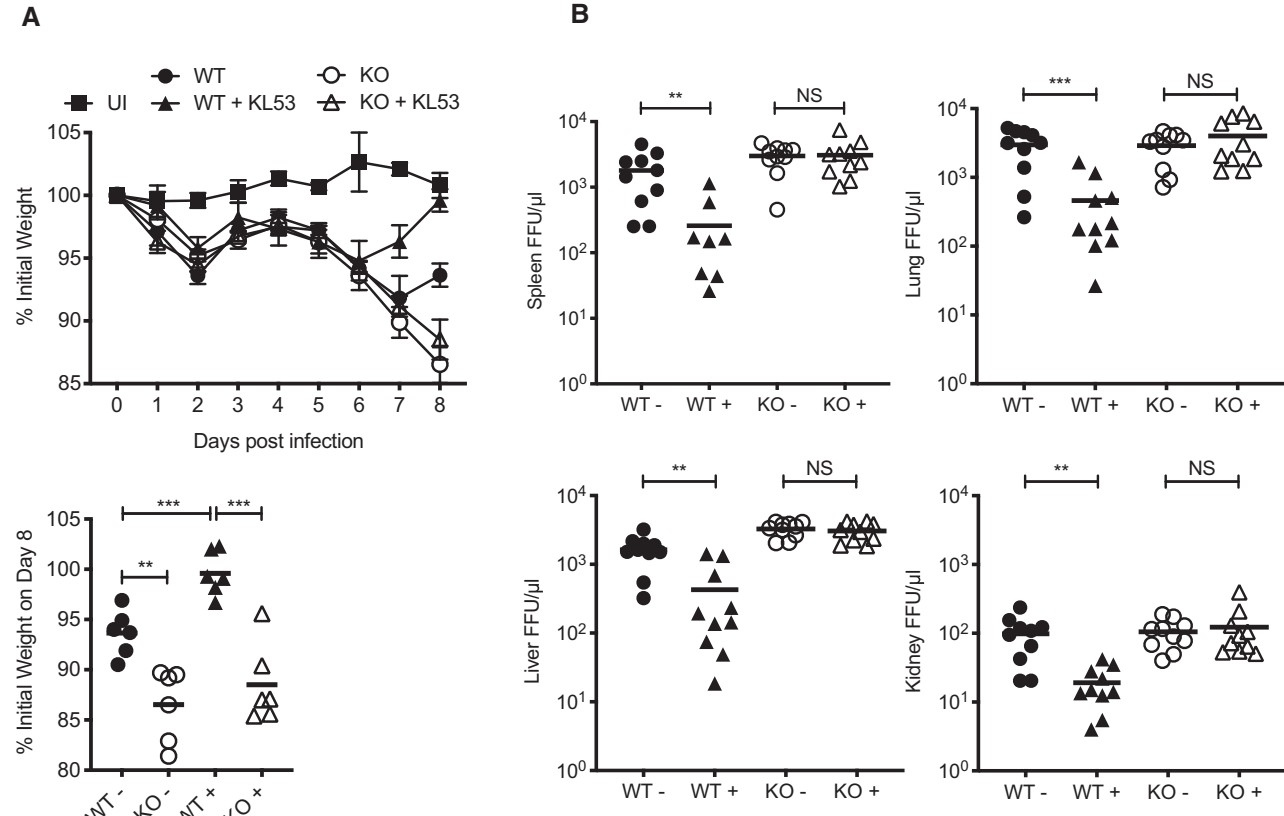

**Figure 2. Non-neutralising anti-N antibodies reduce viraemia but require TRIM21.**

A, B Wild-type (WT) and TRIM21 knockout (KO) mice were infected with 0.5 × 10⁵ FFU LCMV clone 13 (6 mice per group) and either received anti-N mAb KL53 (+) or control (−) intraperitoneally on days 1 and 3pi. Two mice were uninfected (UI). (A) Weights were monitored throughout infection, with final day 8 weights of individual mice presented separately. (B) Viral titres in the spleen, liver, lung and kidney of all mice were determined by FFA day 8pi. Each data point represents one mouse, with results from two repeat experiments combined.

Data information: All data are presented as mean with standard error, **$P < 0.01$, ***$P < 0.001$, NS not significant, unpaired *t*-test.

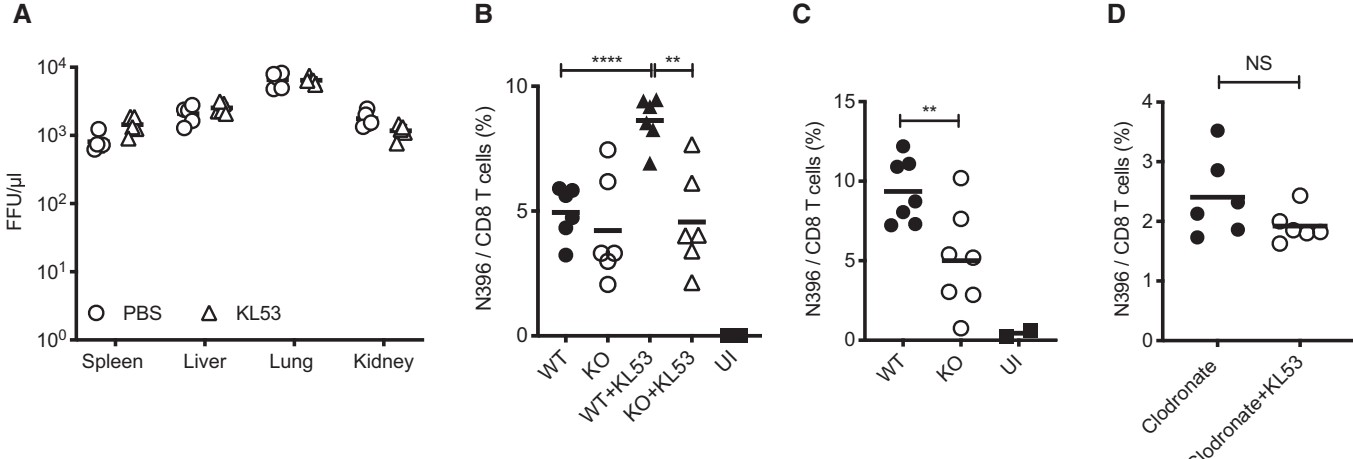

**Figure 3. TRIM21 uses anti-N antibodies to induce an N-specific CTL response.**

A  CD8 T cells in WT mice were depleted by administration of anti-CD8 mAb 1 day prior to infection with $10^5$ FFU LCMV. Anti-N mAb KL53 or PBS control was passively transferred IP on days 1 and 3pi. Viral titres in the spleen, lung, liver and kidney of all mice were determined by FFA day 8pi.

B  Spleens from WT and KO mice +/- mAb KL53 day 8pi with $0.5 \times 10^5$ FFU LCMV were analysed for the presence of LCMV N-specific CTLs by staining with the class I N396-404 tetramer.

C  Spleens from naïve WT and KO mice day 10pi with $0.5 \times 10^5$ FFU LCMV were stained with N tetramer.

D  Macrophages were depleted in WT mice by administration of clodronate liposomes 1 day prior to infection with $0.5 \times 10^5$ FFU LCMV. KL53 was administered on days 1 and 3pi, and N396-specific CTLs in the spleen were measured day 8pi by tetramer staining.

Data information: Each data point represents one mouse from the same experiment. Horizontal bars on each graph correspond to the mean, **$P < 0.01$, ***$P < 0.001$, NS not significant, unpaired *t*-test.

no upregulation in KO mice (Fig 3B). No significant difference was observed in $N_{396}$ CTLs between naïve WT and KO mice. However, there was a significant difference at a later time point (10dpi), confirming that the endogenous antibody response is capable of mediating a comparable increase in $N_{396}$-specific CTLs to passively transferred mAb (Fig 3C). Moreover, the delay in specific CTL induction in naïve WTs compared with those given KL53 correlates with their delay in recovery from weight loss (Figs 1A and 2A). There was no difference in PD-1 expression as a marker of T cell activation/exhaustion on CTLs in any of our experiments (Fig EV1B and C). As TRIM21 is able to efficiently target N immune complexes for degradation (Fig 1H), we considered that this might provide N peptides for presentation to CTLs by antigen-presenting cells (APCs). Dendritic cells have previously been shown to be dispensable during LCMV infection (Hilpert *et al*, 2016), whereas macrophages have been linked to N-antibody enhancement of CTLs against influenza (Laidlaw *et al*, 2013). We therefore targeted macrophages for depletion using clodronate liposomes and re-challenged mice with LCMV in the presence of KL53. Mice treated with clodronate liposomes had fewer F480 + and CD11c + cells and KL53 no longer promoted induction of $N_{396}$-specific CTLs (Figs 3D and EV1C and D).

The above data fit a model in which TRIM21 uses anti-N antibodies to rapidly degrade N protein and promote antigen presentation. This leads to a more effective CTL response, which kills infected cells and clears animals of infection. To test this hypothesis, we performed a series of *in vivo* target cell killing experiments, in which we directly compared the ability of CTLs raised during LCMV infection to kill cells displaying $N_{396}$ peptide. We took splenocytes from CD45.1 WT mice and loaded them with different concentrations of

$N_{396}$ peptide (Fig EV2A). To distinguish between each cell population, we labelled them with different concentrations of cell trace violet (CTV). We then mixed the cells together 1:1:1:1 and transferred them intravenously into WT or KO CD45.2 mice 8 days post-LCMV infection (Fig EV2A and B). Three hours after transfer, mice were culled and the number of CD45.1 CTV-labelled cells was quantified by flow cytometry. Similar numbers of CD45.1 cells were recovered from uninfected mice irrespective of their level of $N_{396}$ peptide presentation (Fig 4A). However, in WT infected mice, there was clear evidence of dose-dependent cell killing, with cells loaded with the highest concentration of $N_{396}$ peptide having the lowest survival. Importantly, there was significantly less cell killing of splenocytes recovered from KO mice and this was true at all levels of $N_{396}$ presentation (Fig 4A). We repeated this experiment in the presence of passively transferred KL53 antibody and observed increased levels of cell killing in infected WT animals (Figs 4B and EV2C). In contrast, KL53 did not give a significant increase in cell killing in KOs at any peptide dose. These results show that TRIM21 and anti-N antibodies promote a more potent anti-N CTL killing response.

## Discussion

It has been shown for many viruses, from arenaviruses (Richter & Oxenius, 2013; Straub *et al*, 2013), to influenza viruses (Sambhara *et al*, 2001; Carragher *et al*, 2008; LaMere *et al*, 2011) and coronaviruses (Nakanaga *et al*, 1986; Lecomte *et al*, 1987) that antibodies targeting the N protein can protect against virus challenge *in vivo*. However, as N protein is an internal antigen, anti-N antibodies cannot prevent virus entry into cells and are typically non-

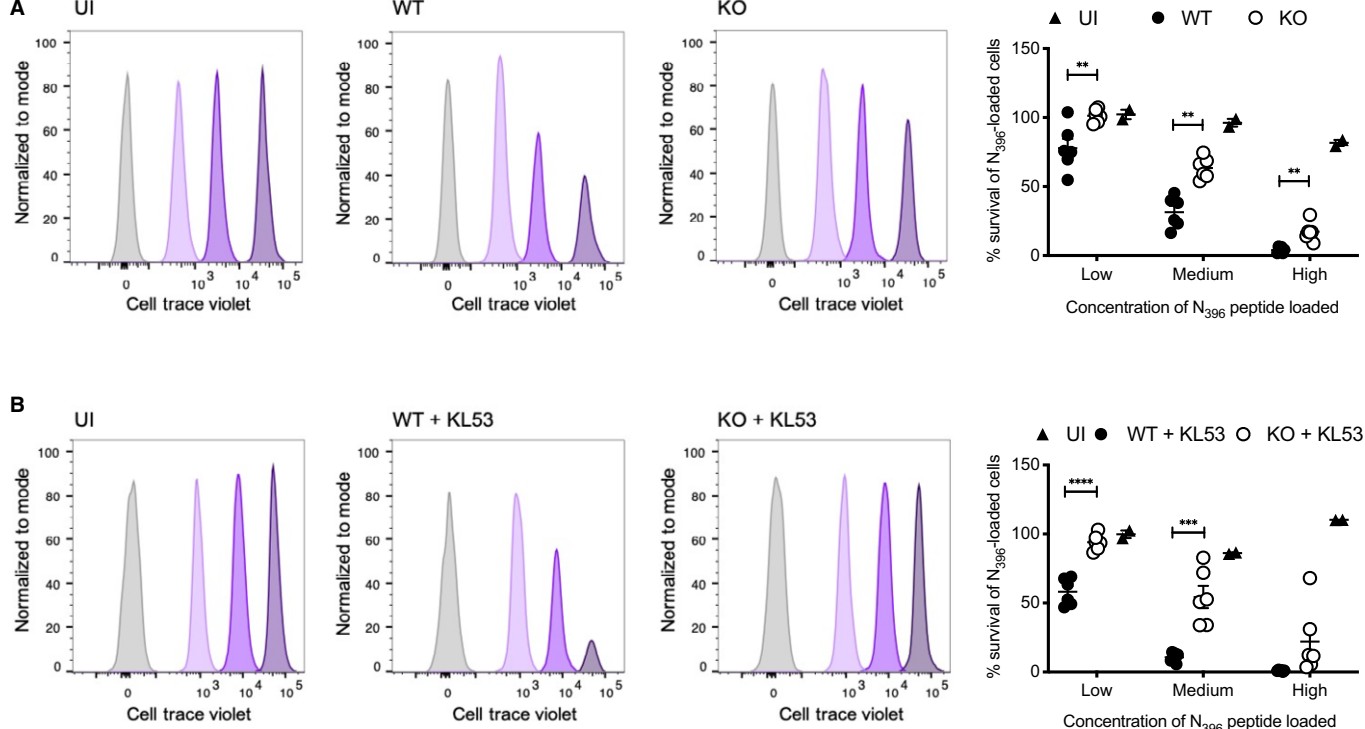

**Figure 4. N-specific CTLs induced by anti-N antibodies and TRIM21 drive potent *in vivo* cell killing.**

A   Splenocytes from uninfected CD45.1 mice, either pulsed with 3 concentrations of N peptide and cell trace violet (CTV) or unlabelled control cells, were transfused intravenously into WT and KO mice (CD45.2) that had been infected with $0.5 \times 10^5$ FFU LCMV 8 days earlier. After 3 h, spleens from recipient mice were harvested and the proportion of CTV-labelled CD45.1 cells was analysed by flow cytometry. Histograms from single representative uninfected (UI), WT and KO mice are presented, showing the proportion of CD45.1 cells remaining for each of the labelled fractions normalised to mode. Summary data from all individual mice in the same experiment are presented in associated scatter plot, showing the mean $\pm$ standard error.

B   Labelled splenocytes as for (A) were transfused into WT and KO mice that had been infected with LCMV 8 days earlier and received mAb KL53 on days 1 and 3pi. Flow cytometry histograms from single representative mice of each genotype. Summary data from all mice in the experiment are presented, showing the mean $\pm$ standard error.

Data information: Horizontal bars on each graph correspond to the mean $\pm$ standard error, **$P < 0.01$, ***$P < 0.001$, unpaired $t$-test.

neutralising. It has also been demonstrated for several viruses, including influenza (LaMere *et al*, 2011) and LCMV (Bergthaler *et al*, 2009), that protection does not require FcγR receptors. Therefore, the mechanisms by which anti-N antibodies mediate protection have remained a mystery.

We have discovered an effector mechanism in which anti-N antibodies activate the cytosolic antibody receptor TRIM21 to mediate protection against viral infection. On the basis of our results, we propose the following model (Fig 5): during infection, antibodies are generated against viral proteins including internal antigens like N protein. This occurs as N protein can be displayed on the surface or released from virally infected cells (Yewdell *et al*, 1981; Lecomte *et al*, 1987; Kyburz *et al*, 1993). This allows the formation of immune complexes that can be taken up by APCs and imported into the cytosol as part of cross-presentation. Once in the cytosol, these immune complexes are recognised as foreign and distinguished from the thousands of resident host proteins by the cytosolic antibody receptor TRIM21. Crucially, TRIM21 is also a ubiquitin ligase and mediates the rapid proteasomal degradation of its targets (Mallery *et al*, 2010; Clift *et al*, 2017). This provides a highly efficient system for generating the peptides necessary for MHC class I

display and drives the stimulation of N protein-specific CD8 T cells. Our present study does not directly address secondary immunity but on the basis of our findings we speculate that the same mechanism of TRIM21-mediated antigen presentation is involved in the restimulation of memory T cells.

While our data demonstrate a role for TRIM21 in antigen presentation, key parts of the process remain unclear. The mechanisms of both endocytic uptake and cytosolic import of exogenous antigens are not currently understood. APCs can process both soluble and cell-associated antigen and utilise multiple mechanisms for uptake, although presentation efficiency appears to be unrelated to the route of entry (Schnorrer *et al*, 2006). Multiple models have been proposed for how antigens and immune complexes gain access to the cytosol, from membrane disruption and leakage into the cytosol (Reis e Sousa & Germain, 1995) to active mechanisms of import requiring the channel forming protein Sec61 (Mukai *et al*, 2011) and the ATPase VCP (Ackerman *et al*, 2006). Cytosolic import is likely to be tightly regulated but the stimuli that control this process are unknown (Kozik *et al*, 2020). The cytosolic import of soluble proteins has been directly demonstrated *in vitro* using β-lactamase-based fluorescence assays (Segura *et al*, 2013) and *in vivo* by

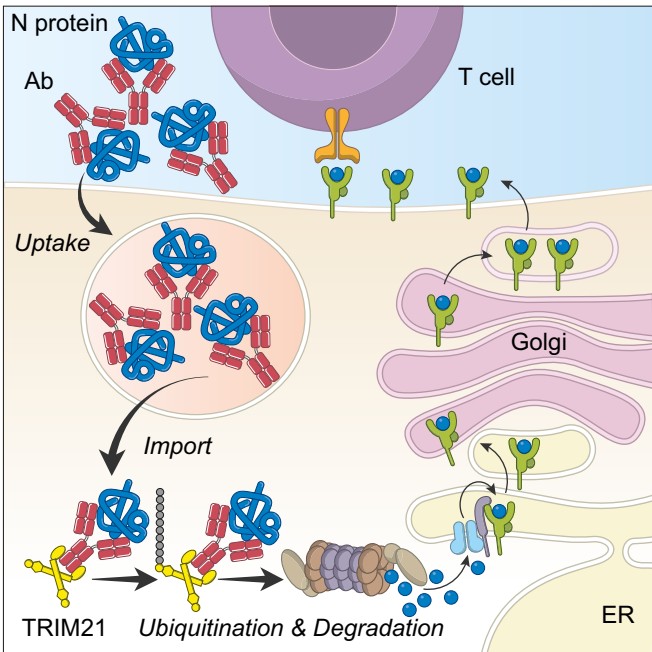

**Figure 5.   Model for TRIM21-mediated Cross-presentation.**

Immune complexes (e.g. Ab:N protein) are phagocytosed—"Uptake"—by APCs, before being moved from the endosome to the cytosol—"Import". While multiple routes for uptake have been described, the mechanism and regulation of exogenous protein import are not well understood. Once in the cytosol, immune complexes are detected by the high-affinity Fc receptor and E3 ubiquitin ligase TRIM21. TRIM21 catalyses ubiquitination to recruit the proteasome and cause complex degradation. Proteasome-derived peptides can then be loaded onto MHC Class I molecules and exported to the cell surface to activate T cells.

injecting cytochrome c into mice and observing selective Apaf-1-dependent cDC1 apoptosis (Lin *et al*, 2008). The importance of cytosolic import during cross-presentation is also inferred from the requirement for cytosolic proteins and machinery. The proteasome in particular is central to cross-presentation, as it is required to generate the peptides for MHC class I display. In contrast, MHC class II peptides are thought to be generated by cathepsins in endolysosomal compartments, making it unlikely that TRIM21 is involved in class II presentation. Selective proteasome inhibitors have been shown to block MHC class I presentation of peptides from both influenza and Sendai virus nucleoprotein but not when the same eptiopes are expressed directly inside the cell as oligopeptides (Mo *et al*, 1999). Moreover, knockout of the immunoproteasome LMP7 prevents peptide presentation from exogenous protein (Palmowski *et al*, 2006). Proteasomes are typically recruited following substrate ubiquitination by an E3 ligase. To our knowledge, TRIM21 is the first E3 ligase capable of proteasomal recruitment that has been identified in cross-presentation. However, TRIM21 is likely to be restricted to immune complex processing because it is dependent upon antibody binding. A single point mutation in IgG, H433A, has been demonstrated to prevent TRIM21 interaction (Foss *et al*, 2016) and abolish its function: Antibodies with mutation H433A lose TRIM21 antiviral activity *in vitro*, in both cell lines (McEwan *et al*, 2012) and primary human macrophages (Labzin *et al*, 2019), and

*in vivo* in a mouse model of infection (Bottermann *et al*, 2018). We show here that an H433A mutant of anti-N antibody KL53 loses the ability to induce TRIM21-mediated intracellular degradation of LCMV nucleoprotein. This is consistent with a requirement for direct antibody interaction in order for TRIM21 to mediate proteasomal degradation of nucleoprotein and enhance MHC class I presentation.

Our experiments have focused on the highly tractable model system provided by LCMV but the similarities in N-antibody protection reported for other viruses suggest that TRIM21 activity is not specific to LCMV but provides a general mechanism to connect the two arms of adaptive immunity and promote synergy between antibodies and T cells. For example, it has been shown that clearance of influenza virus in mice can be promoted by non-neutralising antibodies (Sambhara *et al*, 2001; Carragher *et al*, 2008) and that this correlates with enhanced N-specific CTL responses (LaMere *et al*, 2011). Moreover, it has also been demonstrated that the apparent cooperation between non-neutralising antibodies and CTLs when inducing protective immunity is dependent on the presence of alveolar macrophages (Laidlaw *et al*, 2013). These findings are consistent with our model of anti-N antibodies enhancing antigen presentation via TRIM21 to drive more robust CTL responses. Work to test this hypothesis in the context of influenza infections is currently ongoing.

The existence of an effector mechanism that allows anti-N antibodies to induce cytotoxic T cells has direct implications for ongoing efforts to understand the immune response to SARS-CoV-2. Antibodies are readily generated to target the viral N protein after infection in humans (Liu *et al*, 2020; Sun *et al*, 2020), and preliminary evidence suggests these may be immunodominant (Hachim *et al*, 2020). At present, it is uncertain whether N-specific antibodies are protective against CoV-2, but for murine coronavirus it has been demonstrated that N-specific antibodies can protect against infection *in vivo* (Nakanaga *et al*, 1986; Lecomte *et al*, 1987). This suggests that CoV-2 anti-N antibodies may indeed be protective and not just a marker of prior infection. The identification of anti-N antibodies as a correlate of protection would have particular significance given that the majority of CoV-2 vaccine approaches are currently focusing on the spike protein. We predict that CoV-2 anti-N antibodies will be involved in promoting a protective CTL cell response, and therefore our results support the construction of vaccines based upon N proteins as well as glycoproteins. Furthermore, N proteins are much more highly conserved than surface proteins, which means that immune responses directed against this target are likely to be of greater relevance as the virus evolves.

In summary, we have discovered a missing effector mechanism that explains how non-neutralising antibodies can protect against virus infections *in vivo*. The existence of this mechanism means that TRIM21 has wider role in antiviral immunity than previously thought. It also highlights that antibodies and CTLs are synergistic in their immune protection, which has implications for future vaccines.

# Materials and Methods

### Cells and viruses

Immortalised murine embryonic fibroblasts (MEFs) from wild-type (WT) and TRIM21 deficient (KO) mice were maintained in DMEM

supplemented with 10% foetal calf serum (FCS) and 100 IU/ml penicillin and 100 mg/ml streptomycin. The Clone 13 strain of LCMV was prepared in baby hamster kidney (BHK-21) cells, and viral titre was determined through fluorescent focus assays (FFA) on MEF cells.

## Mice

Seven- to ten-week-old WT C57BL/6 and TRIM21-deficient (T21KO) mice were used in infection experiments, which were conducted in accordance with the 19.b.7 moderate severity limit protocol and Home Office Animals (Scientific Procedures) Act (1986) and approved by the Medical Research Council Animal Welfare and Ethical Review Body. Mice were infected with $0.5^{-1} \times 10^5$ FFU of LCMV by intravenous injection. Administration of LCMV N-specific mAb KL53 (Zeller *et al*, 1988) purified from hybridoma supernatant was performed intraperitoneally for a subset of animals. Throughout the infection protocol, animals were weighed and observed twice daily for clinical signs of infection. Animals that reached the end of the experiment, lost more than 20% of initial body weight or showed moderate clinical signs were culled. Serum samples were collected from the tail vein pre-mortem, and by cardiac puncture post-mortem. LCMV titres were obtained via homogenisation of spleen, kidney, lung and liver, followed by FFA as previously described (Battegay *et al*, 1991).

## *In vitro* neutralisation assays

To deliver antibodies directly to the cytoplasm, antibody or serum at a range of concentrations was electroporated into cells suspended in Neon® Resuspension buffer R using the Neon® Transfection System (Thermo Fisher Scientific), using 2 pulses of 1400 V, 20 pulse width. To observe protein degradation, a "Trim-Away" experiment was performed as described (Clift *et al*, 2017). Recombinant N protein (West *et al*, 2014) and recombinant anti-N antibody were electroporated into WT and T21KO cells, and then 3 h later cells were lysed and probed by Western blot for N protein. Generation of recombinant mAbs was achieved by sequencing the variable regions of KL53, synthesising cDNA encoding the heavy and light chains (Genscript Inc, USA) followed by subcloning into the pFUSE antibody expression vector (Invivogen, USA) in frame with cDNA encoding WT human IgG1 constant region or the H433A mutant.

For intracellular virus neutralisation, electroporated cells were resuspended in antibiotic-free media containing 10% FCS, before being added to wells of a 96 well plate. Cells were incubated at 37°C for 4 h to become adherent to the plate and then washed once with PBS to remove any extracellular antibody. Next, 1,000 focus forming units (FFU) LCMV were added to each well, and infection was allowed to proceed for 16 h at 37°C. Virus was quantified in each well by FFA (Battegay *et al*, 1991).

For extracellular virus neutralisation, serial dilutions of antibodies in complete medium were incubated with 1,000 FFU LCMV in a 1:1 mixture for 1 h at 37°C. The antibody–virus mixture was then diluted 1:10 in complete medium and added in triplicate to MEF cells seeded into a 96-well plate. Infection then proceeded for 16 h at 37°C.

## Serum ELISAs

Ninety-six-well microtitre plates were coated overnight at 4°C with 3 μg/ml purified viral proteins in carbonate/bicarbonate buffer. Plates were washed three times with 0.05% Tween 20 in phosphate-buffered saline (PBS-T) before blocking with 5% skimmed milk-PBS-T. Plates were then incubated for 1 h at 37°C with dilutions of each serum sample in duplicate. After washing, horseradish peroxidase (HRP)-conjugated anti-mouse IgG antibody was added to each well and incubated at 37°C for 1 h. Bound antibody was detected by addition of tetramethylbenzidine (TMB, Invitrogen). The reaction was stopped with 1M $H_2SO_4$, and the optical density (OD) was read at 450 nm with a microplate reader (PHER-Astar).

## Purification and staining of cells from mouse tissues

Splenocyte suspensions were prepared by passing the spleens directly through a 70 μM filter or by first dissecting them into small pieces and digesting them in RPMI containing 1 mg/ml Liberase (Roche) and 0.2 mg/ml DNase I (Roche) for 30 min at 37°C, prior to filtration. Red blood cells were removed using lysis buffer (Biolegend), and then cells were washed by centrifugation in PBS containing 1% foetal calf serum.

Single-cell suspensions were incubated for 30 min at 4°C with Fc Block (anti CD16/CD32 clone 93, Biolegend) and H-2Db tetramer (N396-404) from the NIH Core Tetramer Facility. Cells were next stained with cell surface markers for 30 min at 4°C. The mAbs used were anti-CD45.1 (clone A20, Tonbo Biosciences), anti-CD45.2 (clone 104, Biolegend), anti-CD8a (clone 53-6.7, BD Pharmingen), anti-CD3 clone (17A2, Biolegend), anti-CD44 (clone IM7, BD Horizon) and anti-PD1 (clone J43, BD Pharmingen). Samples were acquired on an LSRFortessa™ (BD Biosciences). The data were analysed using FlowJo software.

## *In vivo* killing assay

Splenocytes from two naive B6.SJL/J mice (CD45.1) were treated with red blood cell lysis buffer, washed and divided into four equal cell populations. Three groups of cells were incubated with N peptide (396–404) at $2 \times 10^{-10}$ M (low), $1 \times 10^{-9}$ M (mid), or $5 \times 10^{-9}$ M (high), for 30 min at 37°C. Cells were then washed and incubated with 0.03, 0.25 or 2 μM of cell trace violet (CTV) (Invitrogen) at 37°C. Excess CTV was removed by washing. The fourth group of cells was unlabelled and unstained. The four cell populations were combined 1:1:1:1, and $10 \times 10^6$ total cells were adoptively transferred intravenously into day 8 LCMV-infected recipient B6 mice. After 3 h, spleens were harvested from recipient mice. Splenocytes were stained with anti-CD45.1 and CD45.2 mAbs. Specific killing was calculated by dividing the number of CD45.1 cells counted in each CTV-labelled group (low, mid or high), with the number of CTV negative CD45.1 cells.

## *In vivo* depletion of cell populations

For depletion of CD8 T cells, rat anti-mouse CD8 IgG2b (Leinco Technologies) (300 μg) was administered IP the day prior to

infection. Control mice received 300 μg of rat isotype control IgG2b (Leinco Technologies). For depletion of macrophages, 1mg clodronate liposomes (Liposoma research liposomes) were administered to mice IV, 1 day prior to infection.

## Data availability

This study includes no data deposited in external repositories.

### Acknowledgements

We thank Erica Ollman Sapphire, Kathryn Hastie and Alessandro Pedroli for contribution of reagents and Patrycja Kozik for helpful discussions. We also thank all at Ares Quarantine, Meng Wang for assistance with mouse experiments and Pablo Rodriguez and Noe Rodriguez for assistance with flow cytometry. We also thank our funders: MRC (UK; U105181010), a Wellcome Trust Investigator Award (200594/Z/16/Z) to LCJ, a Wellcome Trust Clinical Research Career Development Fellowship to SLC, the Research Council of Norway through its Centre of Excellence funding scheme, (Project 179573; JTA and SF), the grants 230526/F20 (JTA) and 251037/F20 (SF), and the South-Eastern Norway project 40018 (JTA). Thanks also to Joanna Westmoreland for preparing graphics.

### Author contributions

Conceptualisation SLC, AO and LCJ.; Formal Analysis, SLC, AO and LCJ.; Investigation, SLC, MV, GP, MW, KO and SF; Resources, DS, JTA, OA; Writing SLC and LCJ.; Funding Acquisition, SLC, JTA and LCJ.

### Conflict of interest

The authors declare that they have no conflict of interest.

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
