## [Review Process File · The EMBO Journal]

Viral nucleoprotein antibodies activate TRIM21 and induce T cell immunity

Sarah Caddy, Marina Vaysburd, Guido Papa, Mark Wing, Kevin O'Connell, Diana Stoycheva, Stian Foss, Jan Terje Andersen, Annette Oxenius, and Leo James

DOI: [10.15252/embj.2020106228](https://doi.org/10.15252/embj.2020106228)

Corresponding author(s): Leo James (lcj@mrc-lmb.cam.ac.uk), Sarah Caddy (slc50@cam.ac.uk)

Review Timeline:

Submission Date:	13th Jul 20
Editorial Decision:	8th Sep 20
Revision Received:	28th Sep 20
Editorial Decision:	6th Oct 20
Revision Received:	15th Oct 20
Accepted:	26th Oct 20

Editor: Karin Dumstrei

Transaction Report:

Dear Leo,

Thank you for submitting your manuscript to The EMBO Journal and also apologise for the delay in getting back to you with a decision. I have now received the three reports on your study that I have enclosed below.

As you can see from the comments, the referees find the analysis very interesting, valuable and very well done. They all support publication here. It makes me very happy to see such positive comments from three good experts in the field. They raise relative minor concerns with the study and most can be addressed with a better discussion and text changes. I like the suggestions provided by referee #2 - point 2 and 3. Point # 3 will also go towards addressing the concern raised by referee #1. Do you have any data on hand to address those points? Would be good to discuss this further and we can do so via email or a video call.

Thank you for the opportunity to consider your work for publication. I look forward to discussing the revisions further with you

best Karin

Karin Dumstrei, PhD
Senior Editor
The EMBO Journal

When assembling figures, please refer to our figure preparation guideline in order to ensure proper formatting and readability in print as well as on screen:
<http://bit.ly/EMBOPressFigurePreparationGuideline>

Further information is available in our Guide For Authors:

The revision must be submitted online within 90 days; please click on the link below to submit the revision online before 7th Dec 2020.

Referee #1:

Manuscript from Caddy et al. describes a mechanism by which anti-Nucleoprotein Abs against LCMV may inhibit virus replication. This mechanism involves TRIM21 FcR intracellular binding of virus Ab immune complexes that triggers efficient TCTL response. Authors use either in vitro transfection of Abs in cells or in vivo treatment of mouse KO for TRIM21. The results described are properly conducted and highly convincing. According to their results, authors proposed an additional model for virus inhibition linking Ab and T cell response.

However, they are some limitation of the experiments proposed. For example, in vitro transfection of the Abs bypass all the first steps of Ab virus entry into the cytosol. If macrophages are doing the jobs as proposed by in vivo experiments, co-culture with macrophages may be envisaged.

Also, it was proposed that Ab/ TRIM21 interaction is dependent on both Ab Fc domain and TRIM21 polymorphism. The role of this interaction was not analyzed by using different Abs, modifying Fc domain or TRIM21 polymorphism. The role of this interaction should at least be discussed.

Authors may design a schematically representation to depict the mechanistic mode proposed of Ab/TRIM21 inhibitory activity. A black box on how Ab enter into cytosol should be symbolized.

Referee #2:

Caddy et al present an interesting study that demonstrates, at least in the LCMV mouse model, that non neutralizing antibodies against nucleocapsid provide some levels of protection in vivo by promoting rapid activation of specific CTLs in a TRIM21 recognition dependent manner, contributing to viral clearance. The authors conduct appropriate mechanistic studies that lead to this conclusion. These observations put together pieces of data that are consistent with what we previously know and offer a well-supported mechanism to explain the "mysterious" ability of nucleocapsid antibodies to provide protection in vivo.

Specific questions

1. One still wonders how important N antibodies are in the presence of existing CD8 memory, as all the studies are conducted during primary infections and therefore, the contribution of N antibodies to protection in the presence of T cell memory is not very clear.

2. The authors might consider conducting loss of function experiments to strengthen the conclusions. For example, by using a virus that has a point mutation in N that prevents binding to the N antibody used in the antibody passive immunization experiments, they could clearly demonstrate that binding of the antibody to N is required for the protection. Alternatively, they might show that TRIM21 mediated protection is compromised in the context of mice lacking B cells unable to generate N antibodies.

3. The use of Fab control antibody in the antibody transfer experiments will confirm that the TRIM21-mediated effect is through recognition of the Fc portion of the antibody-antigen complexes.

Referee #3:

In a beautifully written manuscript the authors show convincingly that antibodies directed against N protein of CMV which are non-neutralizing promote viral clearance by another mechanism. Anti-N antibodies protect the host because they engage cytosolic TRIM21 which facilitates degradation of the attached N protein and priming of LCMV-specific T cells.

1. As a minor suggestion the authors may consider explaining better how they think the immune-complexes might be internalized and whether or not MHC class II also presents N-derived peptides. This comment does not suggest new experiments but would provide a more integrated view of the development of the protective cellular immune response to LCMV.

2. In figure 2A the WT weight is almost 10% higher than KO weight while in figure 1 the two points perfectly overlap. A brief explanation would be helpful.

Referee #1:

Manuscript from Caddy et al. describes a mechanism by which anti-Nucleoprotein Abs against LCMV may inhibit virus replication. This mechanism involves TRIM21 FcR intracellular binding of virus Ab immune complexes that triggers efficient TCTL response. Authors use either in vitro transfection of Abs in cells or in vivo treatment of mouse KO for TRIM21. The results described are properly conducted and highly convincing. According to their results, authors proposed an additional model for virus inhibition linking Ab and T cell response. However, there are some limitations of the experiments proposed. For example, in vitro transfection of the Abs bypass all the first steps of Ab virus entry into the cytosol. If macrophages are doing the jobs as proposed by in vivo experiments, co-culture with macrophages may be envisaged. Also, it was proposed that Ab/TRIM21 interaction is dependent on both Ab Fc domain and TRIM21 polymorphism. The role of this interaction was not analyzed by using different Abs, modifying Fc domain or TRIM21 polymorphism. The role of this interaction should at least be discussed. Authors may design a schematic representation to depict the mechanistic mode proposed of Ab/TRIM21 inhibitory activity. A black box on how Ab enter into cytosol should be symbolized.

We thank the reviewer for their comments. The mechanism of cytosolic import in antigen presenting cells (APCs) is poorly understood. Recent work suggests that import is an inefficient process and APCs require as yet uncharacterised stimulus (Kozik et al., 2020). We are currently working towards an *in vitro* model that we can use to investigate this mechanism further. While the details of IgG:TRIM21 interaction are not investigated in detail in the current study, we have dissected this in previous work. Previously we have shown that TRIM21 has broad antibody specificity, interacting with all IgG subtypes and IgM (Mallery et al., 2010). TRIM21 is highly conserved and human polymorphisms are restricted to rare variants; a recent study based on empirical testing of all variants identified in the 1000 genomes collection concluded that complete loss-of-function would only be predicted in ~1 in a billion individuals (Zeng, Slodkowitz, & James, 2019). TRIM21 is also highly conserved between mammals, with mouse TRIM21 binding human IgG and vice-versa (Keeble, Khan, Forster, & James, 2008). TRIM21:IgG interaction has been characterised in detail by x-ray crystallography (James, Keeble, Khan, Rhodes, & Trowsdale, 2007) and a single IgG point mutation, H433A, is sufficient to prevent interaction (Foss et al., 2016) and specifically abolish TRIM21 function: Antibodies with mutation H433A lose TRIM21 antiviral activity *in vitro*, in both cell lines (McEwan et al., 2012) and primary human macrophages (Labzin et al., 2019), and *in vivo* in a mouse model of infection (Bottermann et al., 2018). Importantly, in the present study, we show that an H433A mutation prevents anti-N antibody KL53 from inducing TRIM21-mediated intracellular degradation of LCMV nucleoprotein. This is consistent with a direct interaction between TRIM21 and IgG being required to generate nucleoprotein peptides for MHC Class I presentation.

We have added new text into the discussion (Lines 222-231) and, as suggested, a schematic giving an overview of our model for TRIM21s involvement in antigen presentation.

Referee #2:

Caddy et al present an interesting study that demonstrates, at least in the LCMV mouse model, that non neutralizing antibodies against nucleocapsid provide some levels of protection in vivo by promoting rapid activation of specific CTLs in a TRIM21 recognition dependent manner, contributing to viral clearance. The authors conduct appropriate mechanistic studies that lead to this conclusion. These observations put together pieces of data that we consistent with what we previously know and offer a well-supported mechanism to explain the "mysterious" ability of nucleocapsid antibodies to provide protection in vivo.

Specific questions

Q1. One still wonders how important N antibodies are in the presence of existing CD8 memory, as all the studies are conducted during primary infections and therefore, the contribution of N antibodies to protection in the presence of T cell memory is not very clear.

A1. This is an important point and, while the present study does not address this directly, on the basis of our findings we speculate that the same mechanism of TRIM21-mediated antigen presentation may help in the restimulation of memory T cells. We have added a sentence to highlight this in the discussion (Lines 193-196).

Q2.1. The authors might consider conducting loss of function experiments to strengthen the conclusions. For example, by using a virus that has a point mutation in N that prevents binding to the N antibody used in the antibody passive immunization experiments, they could clearly demonstrate that binding of the antibody to N is required for the protection.

A2.1 This is a great suggestion and in previous work we have introduced mutations into adenovirus hexon to reduce antibody binding and prevent TRIM21 function (Bottermann et al., 2016). In that case, we had a crystal structure of the antibody:antigen complex. Unfortunately we don't have this information for the binding of KL53 to N and are unable to make a loss-of-binding point mutation.

Q2.2 Alternatively, they might show that TRIM21 mediated protection is compromised in the context of mice lacking B cells unable to generate N antibodies.

A2.2 This is also an excellent suggestion. Previous studies have used a variety of antibody-deficient mouse backgrounds to study LCMV infection. Experiments have been performed in B cell restricted MD4 and T11 μ MT mice, in the B cell deficient JHT strain and in IgMi, which produce little soluble IgG. In each case there is a divergence from wild-type only after the 1st week of infection, with viraemia remaining high. This phenomenon closely matches what we observe in TRIM21 knockouts. We don't currently have double knockout mice in these strains but are establishing them for future work. Our prediction would be that removing TRIM21 from antibody-deficient mice will have no additive effect on LCMV infection, as TRIM21 forms a subset of antibody protection. This prediction is based on previous *in vivo* work where we made a point mutation in a potent antiviral IgG to prevent TRIM21 binding (Bottermann et al., 2018). The mutant antibody no longer blocked adenovirus infection in wild-type mice. In contrast, TRIM21 KO animals were similarly infected whether given unmutated or mutant antibody. These experiments demonstrate that TRIM21 immune protection is dependent upon antibodies.

3. The use of Fab control antibody in the antibody transfer experiments will confirm that the TRIM21-mediated effect is through recognition of the Fc portion of the antibody-antigen complexes.

As described above, we have previously shown in an adenovirus infection model that TRIM21 must interact with the Fc region of IgG in order to mediate its effects *in vivo*. We tested an antibody with mutation H433A, which is located in the Fc, and found that it no longer provided TRIM21-mediated protection. As discussed in detail in the response to reviewer 1, we have extensively characterised this mutant and shown that it specifically abolishes TRIM21 binding and activity. The Fab experiment is a good suggestion, although a Fab may have reduced affinity for LCMV N protein compared to the IgG because it cannot bind bivalently, and this could contribute to reduced protection. It may be possible to clone the KL53 hybridoma, mutate it and produce sufficient recombinant antibody for *in vivo* study and this is something we are actively pursuing.

Referee #3:

In a beautifully written manuscript the authors show convincingly that antibodies directed against N protein of CMV which are non-neutralizing promote viral clearance by another mechanism. Anti-N antibodies protect the host because they engage cytosolic TRIM21 which facilitates degradation of the attached N protein and priming of LCMV-specific T cells.

Q1. As a minor suggestion the authors may consider explaining better how they think the immune-complexes might be internalized and whether or not MHC class II also presents N-derived peptides. This comment does not suggest new experiments but would provide a more integrated view of the development of the protective cellular immune response to LCMV.

A1. We thank the reviewer for their positive comments. We have added a new paragraph into the discussion to explain how immune complexes may be internalized (Lines 198-224), including a sentence on MHC Class II (Lines 213-215)

Q2. In figure 2A the WT weight is almost 10% higher than KO weight while in figure 1 the two points perfectly overlap. A brief explanation would be helpful.

A2. In the experiment in Figure 1, the WT and KO weights diverge at day 9 whereas in the experiment in Figure 2 the weights diverge at day 8. This may be due to slightly different doses being used in these experiments ($10^{4.5}$ FFU for Figure 1 and $0.5 \times 10^{4.5}$ FFU for Figure 2). At the higher dose, the weight loss is slightly steeper and the recovery in WT body weight slightly delayed.

References

- Bottermann, M., Foss, S., van Tienen, L. M., Vaysburd, M., Cruickshank, J., O'Connell, K., ... James, L. C. (2018). TRIM21 mediates antibody inhibition of adenovirus-based gene delivery and vaccination. *Proceedings of the National Academy of Sciences of the United States of America*, 115(41), 10440–10445. <https://doi.org/10.1073/pnas.1806314115>
- Bottermann, M., Lode, H. E., Watkinson, R. E., Foss, S., Sandlie, I., Andersen, J. T., & James, L. C. (2016). Antibody-antigen kinetics constrain intracellular humoral immunity. *Scientific Reports*, 6, 37457. <https://doi.org/10.1038/srep37457>
- Foss, S., Watkinson, R. E., Grevys, A., McAdam, M. B., Bern, M., Høydahl, L. S., ... Andersen, J. T. (2016). TRIM21 Immune Signaling Is More Sensitive to Antibody Affinity Than Its Neutralization Activity. *Journal of Immunology (Baltimore, Md. : 1950)*, 196(8), 3452–3459. <https://doi.org/10.4049/jimmunol.1502601>
- James, L. C., Keeble, A. H., Khan, Z., Rhodes, D. A., & Trowsdale, J. (2007). Structural basis for PRYSPRY-mediated tripartite motif (TRIM) protein function. *Proceedings of the National Academy of Sciences of the United States of America*, 104(15), 6200–6205. <https://doi.org/10.1073/pnas.0609174104>
- Keeble, A. H., Khan, Z., Forster, A., & James, L. C. (2008). TRIM21 is an IgG receptor that is structurally, thermodynamically, and kinetically conserved. *Proceedings of the National Academy of Sciences of the United States of America*, 105(16). <https://doi.org/10.1073/pnas.0800159105>
- Kozik, P., Gros, M., Itzhak, D. N., Joannas, L., Heurtebise-Chrétien, S., Krawczyk, P. A., ... Amigorena, S. (2020). Small Molecule Enhancers of Endosome-to-Cytosol Import Augment Anti-tumor

- Immunity. *Cell Reports*, 32(2), 107905. <https://doi.org/10.1016/j.celrep.2020.107905>
- Labzin, L. I., Bottermann, M., Rodriguez-Silvestre, P., Foss, S., Andersen, J. T., Vaysburd, M., ... James, L. C. (2019). Antibody and DNA sensing pathways converge to activate the inflammasome during primary human macrophage infection. *The EMBO Journal*, 38(21), e101365. <https://doi.org/10.15252/emj.2018101365>
- Mallery, D. L., McEwan, W. A., Bidgood, S. R., Towers, G. J., Johnson, C. M., & James, L. C. (2010). Antibodies mediate intracellular immunity through tripartite motif-containing 21 (TRIM21). *Proceedings of the National Academy of Sciences of the United States of America*, 107(46). <https://doi.org/10.1073/pnas.1014074107>
- McEwan, W. A. ., Hauler, F., Williams, C. R., Bidgood, S. R., Mallery, D. L., Crowther, R. A., & James, L. C. (2012). Regulation of virus neutralization and the persistent fraction by TRIM21. *Journal of Virology*, 86(16). <https://doi.org/10.1128/JVI.00728-12>
- Zeng, J., Slodkowitz, G., & James, L. C. (2019). Rare missense variants in the human cytosolic antibody receptor preserve antiviral function. *ELife*, 8. <https://doi.org/10.7554/eLife.48339>

Dear Leo,

Thanks for submitting your revised manuscript to the EMBO Journal. I have now had a chance to take at the revised version and your response. I appreciate the introduced changes and I am happy to let you know that we will accept the manuscript for publication here.

Before sending you the final accept letter there are just a few last things that we need to resolve.

- Figure 5 is missing
- When you resubmit will you remove the figures from the MS text. As long as we have the figures uploaded as separate files then we are good. The figure legends should be at the very end
- We require a Data Availability section. As far as I can see no data is generated that needs to be deposited in a database. If this is correct please state: This study includes no data deposited in external repositories
- Can you double check that the reference format is OK
- Please also make sure you add the funding info to submission system
- We encourage the publication of source data, particularly for electrophoretic gels and blots, with the aim of making primary data more accessible and transparent to the reader. It would be great if you could provide me with a PDF file per figure that contains the original, uncropped and unprocessed scans of all or key gels used in the figure? The PDF files should be labeled with the appropriate figure/panel number, and should have molecular weight markers; further annotation could be useful but is not essential. The PDF files will be published online with the article as supplementary "Source Data" files.
- We include a synopsis of the paper (see <http://emboj.embopress.org/>). Please provide me with a general summary statement and 3-5 bullet points that capture the key findings of the paper.
- We also need a summary figure for the synopsis. The size should be 550 wide by [200-400] high (pixels). You can also use something from the figures if that is easier.
- I have asked our publisher to do their pre-publication checks on the paper. They will send me the file within the next few days. Please wait to upload the revised version until you have received their comments.

That should be all - let me know if we need to discuss anything further

Best Karin

Karin Dumstrei, PhD
Senior Editor
The EMBO Journal

Further information is available in our Guide For Authors:

The revision must be submitted online within 90 days; please click on the link below to submit the revision online before 4th Jan 2021.

Dear Leo,

Thank you for submitting your revised manuscript. I have now had a chance to take a look at it and all looks good. I am therefore very pleased to accept the manuscript for publication here.

Congratulations on a nice study!

With best wishes

Karin

Karin Dumstrei, PhD
Senior Editor
The EMBO Journal

Please note that it is EMBO Journal policy for the transcript of the editorial process (containing referee reports and your response letter) to be published as an online supplement to each paper. If you do NOT want this, you will need to inform the Editorial Office via email immediately. More information is available here: https://emboj.embopress.org/about#Transparent_Process

Your manuscript will be processed for publication in the journal by EMBO Press. Manuscripts in the PDF and electronic editions of The EMBO Journal will be copy edited, and you will be provided with page proofs prior to publication. Please note that supplementary information is not included in the proofs.

Should you be planning a Press Release on your article, please get in contact with embojournal@wiley.com as early as possible, in order to coordinate publication and release dates.

If you have any questions, please do not hesitate to call or email the Editorial Office. Thank you for your contribution to The EMBO Journal.

Corresponding Author Name: Leo James

Journal Submitted to: The EMBO Journal

Manuscript Number: EMBOJ-2020-106228